# Profiling of metabolites, proteins, and protein phosphorylation in silica-exposed BEAS-2B epithelial cells

Jin Chen[1]*, Hanshi Wang[2], Hongzhi Gao[1], Yiming Zeng[3]

1 Clinical Center for Molecular Diagnosis and Therapy, The Second Affiliated Hospital of Fujian Medical University, Quanzhou, Fujian Province, China, 2 Department of Orthopedic, The Second Affiliated Hospital of Fujian Medical University, Quanzhou, Fujian Province, China, 3 Department of Respiratory Medicine, the Second Affiliated Hospital of Fujian Medical University, Quanzhou, Fujian Province, China

* chenjin@fjmu.edu.cn

## Abstract

Silicosis is an uncurable occupational disease induced by crystalline silica. Increased prevalence of silicosis has resulted in the increased need for development of treatment options. Although macrophages respond first to silica, epithelial cells are also involved in silicosis. However, changes in protein and metabolite levels have not been reported simultaneously. We used mass spectrometry to profile changes in metabolites, proteins, and phosphorylation in silica-exposed BEAS-2B epithelial cells. Silica exposure increased TCA cycle, alanine, aspartate and glutamate metabolism, and aerobic glycolysis activity. In addition, protein levels in the endoplasmic reticulum were significantly altered, and phosphorylation of MAPK signaling proteins was increased. The results of this study increased understanding the role of epithelial cells in silicosis.

## Introduction

Inhalation of free crystalline silica can result in accumulation in the respiratory system, and long-term exposure leads to the occupational disease silicosis [1]. Workers in contemporary practices such as denim sandblasting and the stone industry have experienced increases in silicosis in addition to workers in miners [2]. More than 230 million individuals were exposed to silica, including 2 million in the United States and 23 million in China [3–6]. Although protective measures such as dust control and respirators have been used [1,7,8], the prevalence of the disease remains high worldwide, especially in developing countries. Therefore, additional studies are needed to determine mechanisms by which silicosis occurs to facilitate development of effective interventions.

Macrophages are fundamental to the development of silicosis, and changes in their metabolite and protein levels *in vitro* have been characterized. Specifically, glucose uptake and increased aerobic glycolysis were enhanced, and NF-κB was activated [9,10]. A study showed that epithelial cells were shown to be critical targeted cells in the silicosis progression [11]. They secreted inflammatory factors and signaled to mesenchymal cells, which promoted

**Data Availability Statement:** All the raw files are available from iProx (Project ID: IPX0004776000).

**Funding:** This study is supported by the National Natural Science Foundation of China (81900034) and Natural Science Foundation of Fujian Province

(2021J01255), but the funders had no role in study design, data collection and analysis, decision to publish, or preparation of the manuscript.

**Competing interests:** The authors have declared that no competing interests exist.

**Fig 1. Study design.** (A). Proteomics and phosphoproteomic analysis. (B) Metabolic profiling.

disease progression [12,13]. However, few studies have screened the changes of metabolites, proteins, and phosphorylation in epithelial cells exposed to silica. The human bronchial epithelial cell line, BEAS-2B, has been used to evaluated pulmonary cytotoxicity and dysfunction [14,15]. In this study, we used BEAS-2B cells to investigate silica induced changes.

Mass spectrometry (MS) is a powerful tool for proteomic analysis and metabolite profiling. Global proteomics has been used to characterize silica nanoparticle-induced cytotoxicity in liver cells [16]. Metabolic profiling may be an effective method to illustrate nanomaterial-induced toxicity [17]. A multi-omics approach using transcriptomic, proteomic, and phospho-proteomic profiling showed that silica nanoparticles induced autophagy [18]. However, crystalline silica is different than silica nanoparticles, and its influence on metabolites, proteins, and phosphorylation in epithelial cells required further study.

In this study, we used dimethyl labeling followed by phosphorylated peptide enrichment to investigate the effects of crystalline silica on BEAS-2B cells (Fig 1A). Metabolites such as amino acids, sugars, and organic acids, were analyzed in response to a time course of silica exposure (Fig 1B). The results showed that silica enhanced the TCA cycle, alanine, aspartate and glutamate metabolism, and promoted aerobic glycolysis. In addition, the expression levels of proteins in the endoplasmic reticulum changed significantly, and components of the MAPK signaling pathway were phosphorylated and activated.

## Materials and method

### Cell culture

Human bronchial epithelial cell BEAS-2B were grown according to the supplier's instructions and were cultured in a humidified 5% $CO_2$ atmosphere at 37˚C. The cells were treated with 0, 25, 50, or 100 μg/mL crystalline silica (Min-U-Sill, US Silica Co., Berkeley Springs, WV) for 12, 24 and 48 h.

### Sample preparation for proteomics and phosphoproteomics

The cells were lysed in a buffer containing 50 mM HEPES (pH 7.4), 4% SDS, protease inhibitor cocktail (Sigma, U.S.A.) and phosphatase inhibitors (Beyotime, China), then centrifuged at

18000 g for 20 min at 4˚C to remove debris. Proteins were collected via ice-cold acetone precipitation. The proteins were resuspended in a buffer containing 50 mM HEPES (pH 8) and 8 M urea, and their concentrations were measured using the BCA assay. Then, disulfide bonds were reduced using 20 mM DTT for 2 h at 37˚C, and alkylation was performed using 40 mM IAA. After 8-fold dilution, the samples were incubated with trypsin (1:40, w/w) 16 h to digest proteins. Dimethyl isotopic labeling was conducted according to the previous study [19]. In brief, samples (100 μg) without and with silica treatment were labeled with light (4 μL of $CH_2O$, 4%, v/v) and heavy (4 μL of $CD_2O$, 4%, v/v) dimethyl isotopic reagents, respectively, and incubated with fresh catalyst 4 μL $NaBH_3CN$ (0.6 M). After incubation for 1 h at room temperature, the reaction was quenched by addition of 8 μL formic acid (5%, v/v). The samples were mixed and desalted using a C18 column (Waters, U.S.A.).

Phosphopeptides were enriched using Ti-IMAC (J&K Scientific, China) as we described previously [20]. Briefly, the mixture was incubated with Ti-IMAC in a buffer containing 80% ACN and 6% TFA, then washed with washing buffer 1 (50% ACN, 0.1% TFA, 200 mM NaCl) and washing buffer 2 (30% ACN, 0.1% TFA). Phosphopeptides were eluted using 10% $NH_3 \cdot H_2O$ and lyophilized for LC-MS/MS analysis.

## Metabolite extraction

The cells were added to a 4× volume of methanol containing tridecanoic acid (internal standard; 5 μg/mL) and vortexed for 30 s. Trimethylsilylation was performed as previously described [21]. The supernatant solution was lyophilized and redissolved in 50 μL methoxyamine solution (20 mg/mL in pyridine), vortexed for 1 min, then shaken for 90 min at 30˚C. N-Trimethylsilyl-N-methyl trifluoroacetamide (MSTFA) was then added, and the samples were incubated for 30 min at 37˚C. After centrifugation, the supernatant was used for GC-MS analysis.

## GC-MS analysis

GCMS-QP2020 instrument equipped with an AOC-20i auto-sampler (Shimadzu, Kyoto, Japan) was used in SIM mode and data were acquired using Smart Database software (Shimadzu, Kyoto, Japan). The injection volume was 1 μL with splitless. Reaction products were separated by a DB-5 capillary column (30 m×0.25 mm×1 μm, Agilent Technologies, Palo Alto, CA.) The flow rate of helium (carrier gas) was set at 40 cm/s. Electron impact ionization was used, and the voltage of the detector was set according to the tuning result. All other parameters were set to default.

## HPLC-MS/MS analysis

LTQ-Orbitrap Elite (Thermo Scientific) coupled with Dionex UltiMate 3000 RSLCnano system (Thermo Scientific) was used for analysis. Peptides and phosphopeptides were loaded onto a C18 trap column and separated using an 18 cm× 75 μm i.d. C18 (1.9 μm, 120 Å, purchased from Dr. Maisch, Germany) capillary column at a flow rate of 300 nL/min. The parameters used were described in our previous paper [22]. All MS and MS/MS spectra were acquired in data-dependent analysis mode, in which the 20 most intense ions were selected for MS/MS scan via collision-induced dissociation with 35% normalized collision energy. The dynamic exclusion parameters were as follows: repeat count 1, repeat duration 30 s, and exclusion duration 90 s.

## Data processing

MS data were processed using MaxQuant (Version 1.5.8.3) using Andromeda as a search engine against the Uniprot human protein database (Download from uniport with Proteome ID UP000005640) with precursor mass tolerance of 4.5 ppm and fragment mass deviation of 0.5 Da. Variable modifications consisted of methionine oxidation, acetylation of protein N-term. Fixed modification contained cysteine carbamidomethylation. Light labeled samples were set to dimethyl Lys 0 and N-term 0, while heavy labeled samples were set to dimethyl Lys 4 and N-term 4. Trypsin was set as specific proteolytic enzyme. Peptides with a minimum of seven amino acids and a maximum of two missed cleavages were allowed in the analysis. For peptide and protein identification, the false discovery rate cutoffs were both set to 0.01. Match between runs was selected. All other parameters were set to default values. The mass spectrometry proteomics data have been deposited to the ProteomeXchange Consortium (http://proteomecentral.proteomexchange.org) via the iProX partner repository with the dataset identifier PXD035608.

Metabolite data collected using GC-MS were normalized by internal standard and peak area. Significantly differentially abundant metabolites were determined using ANOVA analysis. SIMCA (version 14.1) was used for PCA analysis.

Volcano, KEGG, and GO analyses were performed using Rstudio. For proteomic analysis, significantly differentially expressed proteins were extracted when their quantified ratios were >1.3 or <0.76 with RSD <0.3 in three experiments. For phosphoproteomic analysis, the RSD cutoff was 0.5.

## RNA extraction and reverse transcription-quantitative polymerase chain reaction (RT-qPCR) analysis

The cells were homogenized, and extracted via chloroform/isopropanol precipitation. The primers sequences were as follows: IL8, 5'- ATG ACT TCC AAG CTG GCC GTG GCT-3' (forward) and 5'-TCT CAG CCC TCT TCA AAA ACT TCT C-3'(reverse); GAPDH, 5'-CCACCCATGGCAAATTCCATGGCA-3' (forward) and 5'-TCTAGACGGCAGGTCAGGTCCACC-3'(reverse). Then, RT-qPCR was performed using SYBR premix EX TaqTM (Takara, Kusatsu, Shiga, Japan) with an Applied Biosystem 7500 Real Time PCR System. The results were quantified using the 2-ΔΔCt method.

## ELISA and CCK8 assay

Secreted IL-8 was detected using ELISA (Lianke Biotech. Co. LTD, Hangzhou, China) according to the manufacturers' instructions. Cell cytotoxicity was measured using Cell Counting Kit-8 (Meilunbio, China) according to the manufacture's instructions.

## Western blots

The cells were lysed in RIPA buffer supplemented with protease and phosphatase inhibitors (Beyotime, China). Following SDS-PAGE and wet transfer onto PVDF membranes, proteins were detected by western blotting with specific antibodies: anti- NF-κB antibody (1:2000); anti-phospho-NF-κB antibody (1:2000); anti-actin antibody (1:2000) (Cell Signal Technology, U.S.A.). The secondary antibody was anti-rabbit IgG (HRP-linked antibody) (Beyotime, China). Signal was detected by an ImageQuant LAS 4000mini system (GE Healthcare, U.S.A.) using ECL reagent (Beyotime, China). Semi-quantitative analyses was performed using Quantity One software (Bio-Rad, the U.S.A.). Moreover, all the blot images were uploaded to the public-available data repository, Figshare, with DOI 10.6084/m9.figshare.20486604.

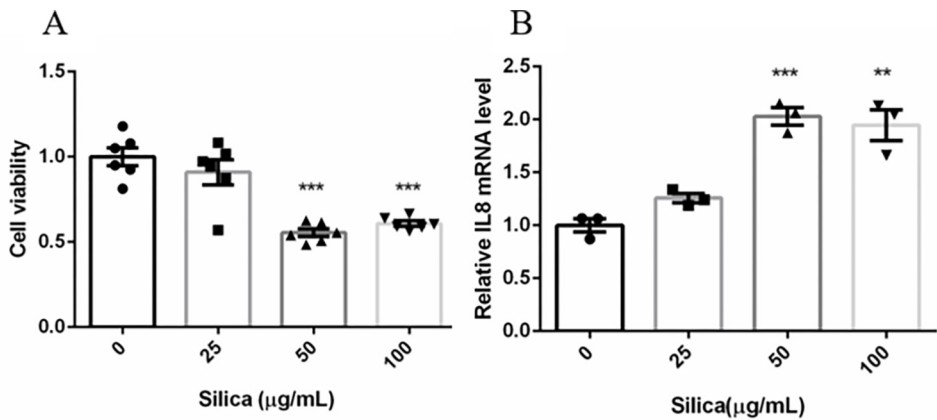

**Fig 2. The influence of crystalline silica on BEAS-2B cells.** (A). Cell viability in response to different concentrations of silica. (B). IL-8 mRNA expression in response to different concentrations of silica. Data are presented as the mean ± SEM; ***, p<0.001; **, p<0.01.

## Results

### The influence of crystalline silica on BEAS-2B cells

To investigate epithelial cell proliferation in response to silica, BEAS-2B cells were treated with 0, 25, 50, or 100 μg/mL crystalline silica for 24 h. Cell viability decreased to nearly 50% in response to 50 and 100 μg/mL crystalline silica (Fig 2A). These results indicated that crystalline silica was toxic to BEAS-2B cells. We then measured the inflammation cytokine IL-8 to further characterize the effects of silica on epithelial cells. The results showed that IL-8 levels were significantly increased in response to 50 and 100 μg/mL silica (Fig 2B). Therefore, we used 50 μg/mL silica for subsequent experiments.

### Proteomic analysis of silica-treated when BEAS-2B cells

BEAS-2B cells were treated with 50 μg/mL crystalline silica for 24 h, and proteins were analyzed. Three biological replicates resulted in quantification 1920, 1821, and 1934 proteins, among which 1636 proteins were common to the three replicates (Fig 3A). We analyzed proteins that were detected in two or more of the replicates, and volcano graphs showed significant proteins with good repeatability (Fig 3B, S1 Table). Proteins that changed significantly in response to silica were analyzed using KEGG pathway and GO analyses (Fig 3C–3F). Spliceosome and ribosome were the top two enriched pathways (Fig 3C), and the significance of both were further verified in cellular component analysis (Fig 3D). Biological function analysis of differentially expressed proteins showed that the expression levels of proteins in the endoplasmic reticulum were significantly influenced by silica (Fig 3E). Molecular function enrichment analysis showed that proteins involved in structural constituent of ribosome, oxidoreductase activity, and protein binding were significantly altered in response to silica (Fig 3F). These results showed that protein expression was influenced by silica exposure.

### Phosphoproteome analysis of silica-treated BEAS-2B cells

Phosphopeptides were extracted and detected following peptides quantification. Analysis of three replicates resulted in quantification of 1830, 1759, and 1555 phosphorylation sites in three replicates (Fig 4A, S2 Table). We selected phosphorylation sites which were quantified in two or more samples, and significant sites were presented in color in a volcano plot (Fig 4B).

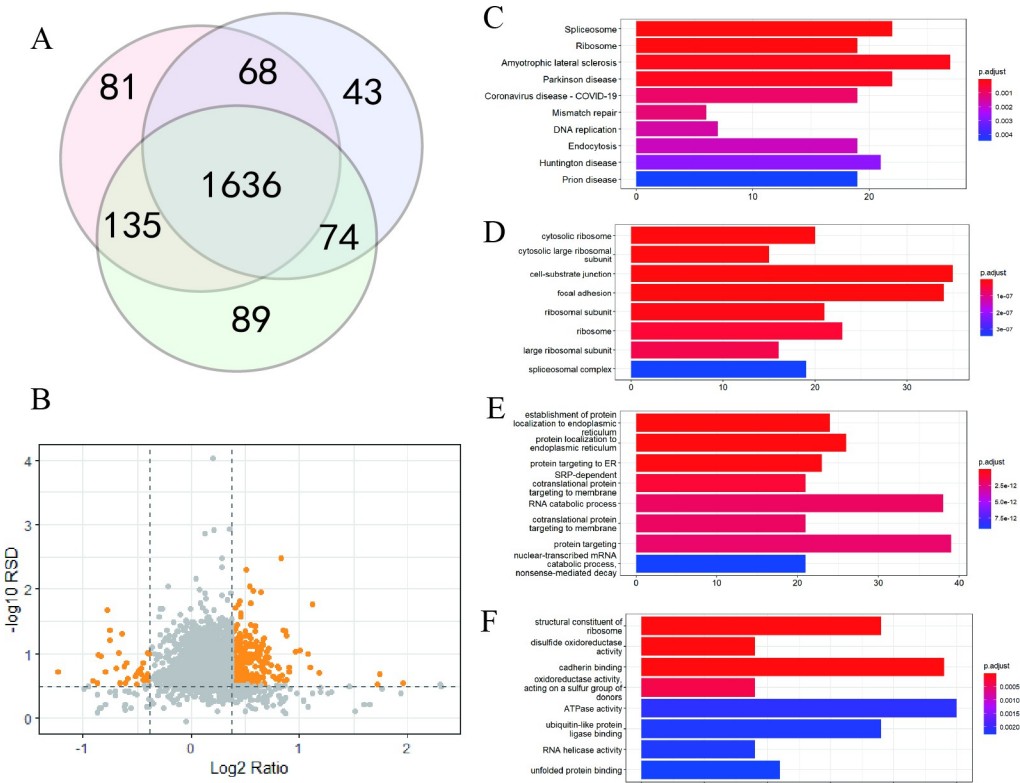

**Fig 3. Proteomic analysis when BEAS-2B treated with silica.** (A) Venn diagram of the quantified proteins. (B) Volcano graph. Significantly differentially expressed proteins are presented in color. (C) KEGG enrichment analysis of differentially expressed proteins. (D-F) GO analysis of differentially expressed proteins, including cellular component (D), biological function (E), and molecular function (F).

We analyzed the enrichment of differentially expressed phospho-proteins via KEGG pathway and GO analyses (Fig 4C–4F). The spliceosome pathway was identified as significant in KEGG analysis (Fig 4C), which was consistent with the proteomics results (Fig 3C). Biological function analysis showed that RNA splicing was significantly affected by silica (Fig 4D). Cell component analysis showed that focal adhesion, cell-substrate junction, and other components were significantly changed in response to silica (Fig 4E). Finally, molecular function analysis described that protein binding showed an important role during silica treatment (Fig 4F).

We then normalized the changes of phosphorylation sites based on their protein levels to profile the alteration of phosphorylation intensity induced by silica (S3 Table). Phosphoproteome analysis showed that the MAPK signaling pathway was significantly affected by silica treatment. As the MAPK pathway was not altered in the proteomic analysis, changes in the phosphoproteome may be better indicators of the effects of silica on signal transduction. We summarized phosphorylation of TP53, TAOK1, STMN1, RRAS2, NFKB1, JUND, JUN, HSPB1, FLNB, FLNA, EGFR, CACNA1B, and CACNA1A in the MAPK signaling pathway (Table 1). Not surprisingly, some quantified proteins changed little (TP53, STMN1, RRAS2, HSPB1, FLNB, FLNA, and EGFR) and the others were not quantified in proteomics analysis (Table 1). Exposure to silica resulted in phosphorylation of NFKB1, which represented activation of NF-κB. We then detected its expression via Western bolt, and exposure to silica for 24 h resulted in the increased NF-κB phosphorylation in the absence of changes in NF-κB protein levels (Fig 5A and 5B). Activation of NF-κB promoted secretion IL-8, which was described

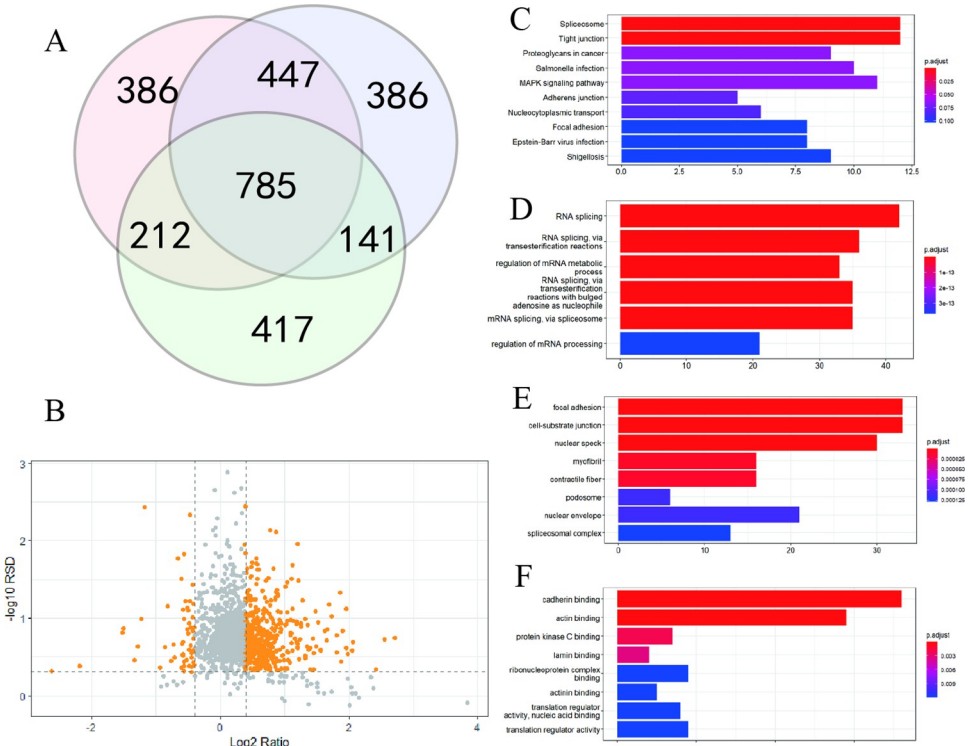

**Fig 4. Phosphoproteomic analysis when BEAS-2B treated with silica.** (A) Venn diagram of the quantified phosphorylation sites. (B) Volcano graph. Significant phosphorylation sites are presented in color. (C) KEGG enrichment of differentially phosphorylated proteins. (D-F) GO analysis of differentially phosphorylated proteins, including biological function (D), cellular component (E), and molecular function (F).

before (Fig 2B). These results showed that the phosphoproteome could provide valuable information regarding silica-induced signaling.

## Metabolic profiling of silica-treated when BEAS-2B cells

We extracted metabolites from BEAS-2B cells treated with silica at 0, 24, and 48 h (6 replicates). The 0, 24, and 48 h silica treatment samples were clearly differentiated through PCA analysis (Fig 6A), which indicated significant changes in metabolites in response to silica treatment. Trimethylsilylation followed by GC-MS analysis resulte in detection of 182 metabolites. Metabolites that were significantly changed in response to silica are shown in the heatmap (Fig 6B, S4 Table). Levels of hydroxyproline, an indicator of fibrosis, were greatly increased with 48 h exposure to silica (Fig 6C). In addition, Taurine and hypotaurine metabolism, alanine, aspartate and glutamate metabolism, and beta-alanine metabolism pathways were enriched in response to silica exposure (Fig 6D).

We extracted significantly differentially abundant proteins and metabolites and found they were enriched in the pathway of TCA cycle, alanine, aspartate and glutamate metabolism, and pyrimidine metabolism (Fig 7A). We generated a figure describing dynamic changes in epithelial cells exposure to silica (Fig 7B). Pyruvic acid, citric acid, aconitic acid, α-ketoglutaric acid, succinic acid, and fumaric acid increased with duration of silica exposure, which indicated that the TCA cycle was affected by silica exposure. Increased levels of alanine, aspartic acid, N-acetylaspartic acid, ureidosuccinic acid, glutamic acid, and asparagine indicated that alanine, aspartate and glutamate metabolism were altered by silica exposure. The expression levels of

**Table 1. Changes of proteins in MAPK signaling pathway.**

| Proteome | | Phosphoproteome | | |
|---|---|---|---|---|
| Gene name | Variation induced by silica (%) | Gene name | Site | Variation induced by silica (%) |
| TP53 | 22% | TP53 | S315 | 62% |
| TAOK1 | unquantified | TAOK1 | S445 | 112% |
| STMN1 | 16% | STMN1 | S46 | 83% |
| | | | S38 | 62% |
| | | | S25 | 209% |
| | | | S16 | 59% |
| RRAS2 | 11% | RRAS2 | T190 | 34% |
| | | | S186 | 33% |
| NFKB1 | unquantified | NFKB1 | S907 | 65% |
| JUND | unquantified | JUND | S251 | 43% |
| | | | S259 | 46% |
| | | | S90 | 69% |
| JUN | unquantified | JUN | S73 | 83% |
| | | | T62 | 164% |
| HSPB1 | 8% | HSPB1 | S15 | 232% |
| FLNB | 10% | FLNB | S2107 | 38% |
| FLNA | 13% | FLNA | T2336 | 31% |
| | | | S1084 | 31% |
| EGFR | 29% | EGFR | S695 | 129% |
| | | | T693 | 148% |
| CACNA1B | unquantified | CACNA1B | Y463 | -56% |
| CACNA1A | unquantified | CACNA1A | S779 | 59% |
| | | | T778 | 59% |

the proteins PDHB, DLAT, SUCLG1, and CAD increased in response to silica exposure. The results indicated silica exposure resulted in increased aerobic glycolysis. Thus, the combination of proteomic and metabolic analysis gave more reliable data for mechanism discussion. In conclusion, we inferred that the mechanism of silica toxicity may include changes in the TCA cycle and alanine, aspartate and glutamate metabolism.

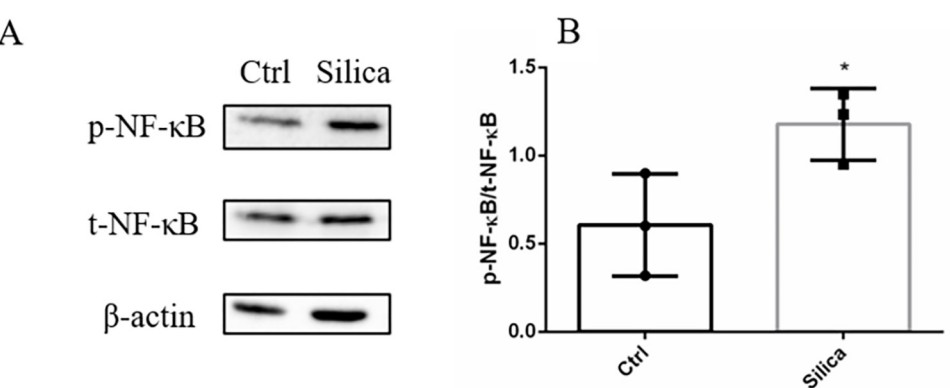

**Fig 5. NF-κB was activated in BEAS-2B cells exposed to silica.** (A). Western blot of NF-κB and its phosphorylation. (B) Quantification of the bands from western blot. Data are presented as the mean ± SEM; *, p<0.05.

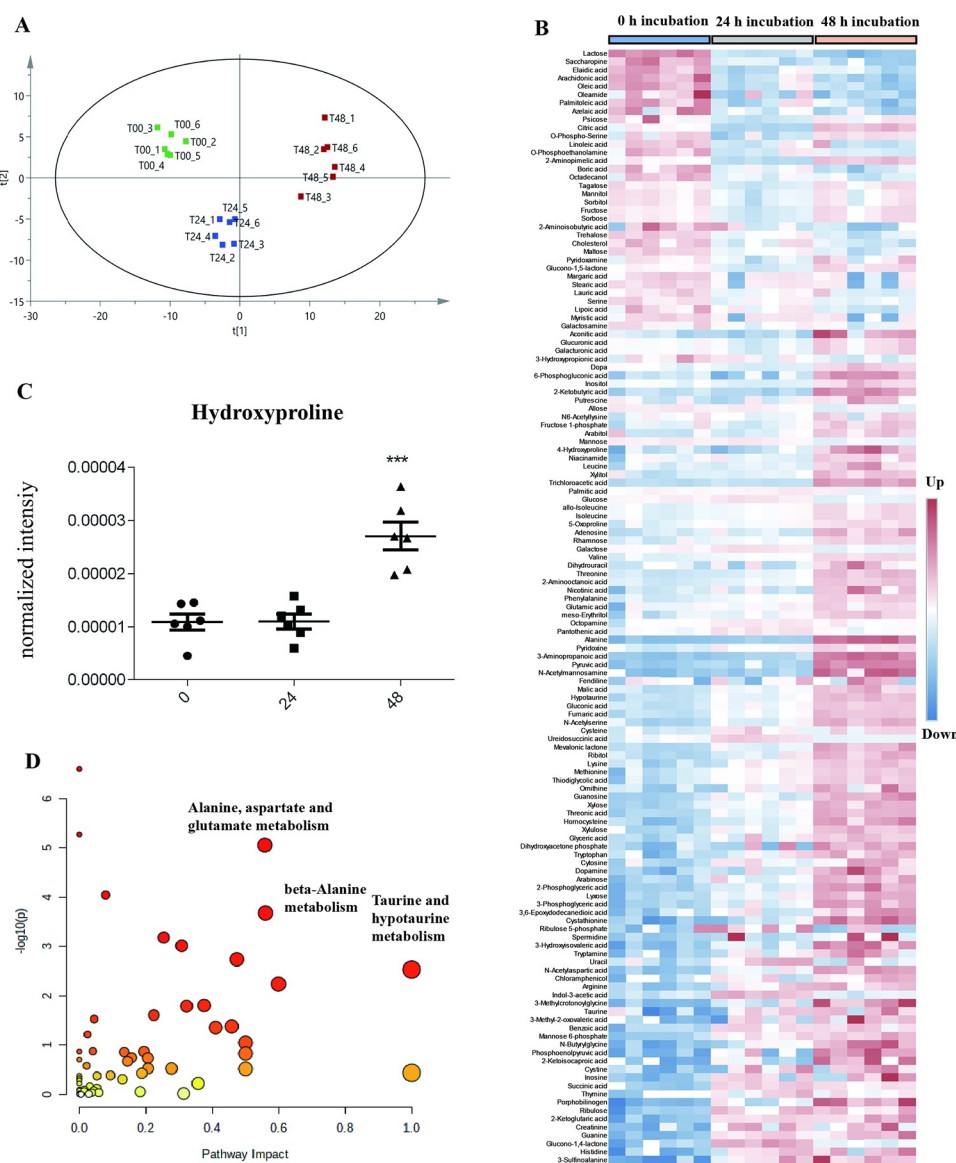

**Fig 6. Metabolic profiling of BEAS-2B treated with silica.** (A) PCA analysis of samples treated with silica for 0, 24, or 48 h. (B) Heatmap of differentially abundant metabolites in response to treatment with silica for 0, 24, or 48 h. (C) Changes in levels of hydroxyproline as determined using GC-MS analysis. Data are presented as the mean ± SEM; ***, p<0.001. (D) Metabolic pathway analysis of differentially abundant metabolites.

## Discussion

Our results showed that silica was toxic to the BEAS-2B epithelial cells and induced significant changes in levels of metabolites, proteins, and phosphorylated proteins. The TCA cycle and alanine, aspartate and glutamate metabolism were significantly affected by silica exposure. In addition, the expression levels of proteins were altered in the endoplasmic reticulum, and the phosphoproteome was significantly affected.

MS-based omics technology has provided valuable information in clinical research. Characterization of the proteome is an excellent strategy to determine key biological process in signaling [23]. In our study, phosphoproteome analysis showed many meaningful changes in phosphorylation. Silica-induced production of cytokines and chemokines in pulmonary

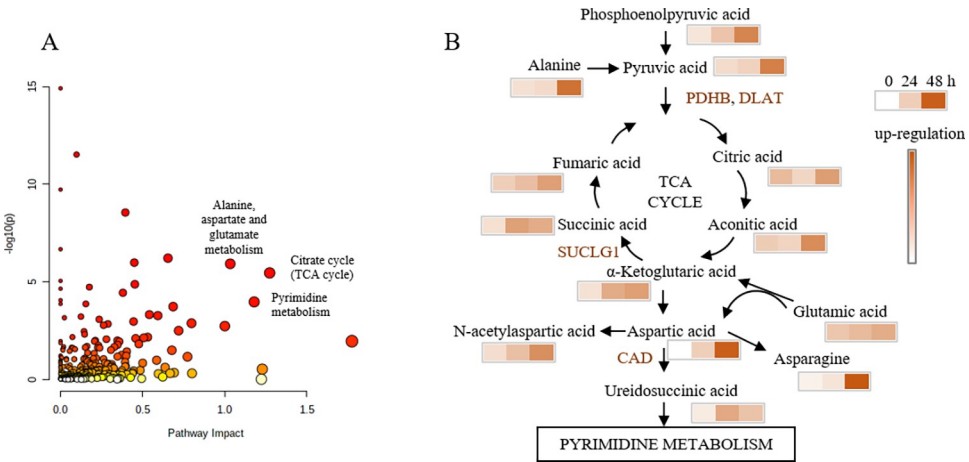

**Fig 7. Pathway analysis using differentially abundant metabolites and differentially expressed proteins.** (A) Enriched KEGG pathway. (B) Metabolites and proteins in the TCA cycle and in alanine, aspartate and glutamate metabolism.

epithelial cells has been shown to occur via NF-κB-dependent and -independent mechanisms *in vivo* and *in vitro* study [24]. NF-κB enhances transcription of inflammatory factors, such as IL-8, to promote inflammation [25,26]. Our results showed that silica exposure resulted in increased phosphorylation of NF-κB, which was indicative of activation and subsequent inflammation. Furthermore, the phosphorylation of serine 727 in STAT3 was significantly increased, which is associated with regulation of transcription of inflammatory cytokines [27]. Moreover, phosphorylation of serine 73 in JUN was enhanced, in which has been previously associated with development of silicosis [28]. These results indicated that the identified phosphorylation of proteins in the MAPK signaling pathway may be a mechanism of silica toxicity.

Metabolic profiling can reveal specific disease targets [29], and may reflect the status of the cell. Hydroxyproline is an indicator of silica-induced fibrosis [30], and was significantly increased in response to 48 h silica exposure in our study. This result indicated that MS-based metabolite profiling could reveal the status of epithelial cells with silica treatment. A previous study showed that aerobic glycolysis in macrophages were strengthened with silica exposure [9]. Our results showed that phosphoenolpyruvic acid, 3-phosphoglyceric acid, 2-phosphoglyceric, and dihydroxyacetone phosphate levels increased with the increased duration of silica exposure, which indicated that silica exposure induced aerobic glycolysis in epithelial cells. These results showed that macrophages and epithelial cells responded in similar manners to silica exposure.

## Conclusion

In conclusion, we investigated crystalline silica-induced changes in metabolites, proteins, and phosphorylation sites in BEAS-2B epithelial cells. The results showed that TCA cycle and alanine, aspartate and glutamate metabolism were increased. The expression levels of proteins in the endoplasmic reticulum were significantly altered, and MAPK protein phosphorylation was significantly altered, which may indicate targets for determinations of mechanisms of silica toxicity. These results provided a landscape for better understanding the role of epithelial cells in silica toxicity. Since silicosis is a progressive lung disease with no known effective therapy, our study may provide insights into the mechanisms of progression of silicosis, and may allow for development of effective interventions.

## Supporting information

**S1 Table. Protein identified in two or more independent experiments.**
(XLSX)

**S2 Table. Phosphorylation sites quantified in two or more independent experiments.**
(XLSX)

**S3 Table. The normalized changes of phosphorylation sites based on their protein levels.**
(XLSX)

**S4 Table. Significant differential metabolites in four or more independent experiments.**
(XLSX)

**S1 Raw image.**
(PDF)

## Acknowledgments

We really appreciate the support from the Second Affiliated Hospital of Fujian Medical University.

## Author Contributions

**Data curation:** Hanshi Wang.

**Formal analysis:** Hanshi Wang.

**Methodology:** Jin Chen.

**Project administration:** Jin Chen.

**Supervision:** Hongzhi Gao, Yiming Zeng.

**Writing – original draft:** Jin Chen.

**Writing – review & editing:** Jin Chen.

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
