## [Decision Letter · Decision Letter 0]

7 Jul 2022

PONE-D-22-15393Profiling of metabolites, proteins, and protein phosphorylation in silica-exposed BEAS-2B epithelial cellsPLOS ONE

Dear Dr. Chen,

Thank you for submitting your manuscript to PLOS ONE. After careful consideration, we feel that it has merit but does not fully meet PLOS ONE’s publication criteria as it currently stands. Therefore, we invite you to submit a revised version of the manuscript that addresses the points raised during the review process.

Please include the following items when submitting your revised manuscript:A rebuttal letter that responds to each point raised by the academic editor and reviewer(s). You should upload this letter as a separate file labeled 'Response to Reviewers'.A marked-up copy of your manuscript that highlights changes made to the original version. You should upload this as a separate file labeled 'Revised Manuscript with Track Changes'.An unmarked version of your revised paper without tracked changes. You should upload this as a separate file labeled 'Manuscript'.

We look forward to receiving your revised manuscript.

Kind regards,

Aleksandra Nita-Lazar, Ph. D.

Academic Editor

PLOS ONE

"This study was supported by the National Natural Science Foundation of China (81900034), Natural Science Foundation of Fujian Province (2021J01255), and the Second Affiliated Hospital of Fujian Medical University (2020BSH03)"

"The authors received no specific funding for this work."

Reviewers' comments:

Reviewer's Responses to Questions

**Comments to the Author**

1. Is the manuscript technically sound, and do the data support the conclusions?

Reviewer #1: Partly

Reviewer #2: Partly

2. Has the statistical analysis been performed appropriately and rigorously? 

Reviewer #1: No

Reviewer #2: I Don't Know

3. Have the authors made all data underlying the findings in their manuscript fully available?

Reviewer #1: Yes

Reviewer #2: No

4. Is the manuscript presented in an intelligible fashion and written in standard English?

Reviewer #1: Yes

Reviewer #2: Yes

5. Review Comments to the Author

Reviewer #1: In this manuscript, Chen et al described a multi-omics approach and results of profiling of metabolites, proteins, and protein phosphorylation in silica-exposed BEAS-2B epithelial cells, in an attempt to understand the biology of Silicosis induced by crystalline silica. Their findings showed metabolites, proteins and phosphorylation levels in many pathways were significantly altered. Many of them confirmed the findings in other similar researches.

In order to deep profile the proteome and phosphopeptidome, fractionation at protein and/or peptide level is a must for mass spectrometry analysis operated in data-dependent acquisition mode. However, this study didn't use any fractionation to spread out the peptides for the mass spec to read. Therefore, the comprehensiveness of this study was largely impaired by the relatively low numbers of proteins and phosphopeptides detected.

Furthermore, normalization is necessary to set the base line for multiple data sets, especially for multi-omics data. This study didn’t clearly state how the data sets were normalized and cross compared. From Line 80 of Sample Prep section, it appeared that loading amounts based on protein concentrations were used to normalize for protein- and phosphopeptide- level comparison. Whereas for metabolites, internal standard (tridecanoic acid, 5 μg/mL) was added to the cells by 4:1 volume ratio (Line 96) for normalization. But knowing that cell viability varied significantly for control and treatment samples, the cell volumes and protein contents/concentration could be assumed to have greatly changed as well, therefore not a good means to normalize on. And for phosphorylation, this study only listed proteins in MAPK signaling pathway in Table 1 to show no or little protein level changes but significant changes at phosphorylation level. It would be nice to see all the phosphopeptides identified to be normalized based on the protein level changes.

Reviewer #2: In the manuscript, the authors describes proteomics, phosphoproteomics, and metabolomics analysis of epithelial cells with and without crystalline silica exposure.

The article is well written and easy to follow. Unfortunately, figures are with low resolution and I can't read details from them. Raw data need to be provided for evaluation too, which is missing. With this I can't make a responsible review of the presented manuscript.

Here are some suggestions from what I can read:

1. in supplemental table, highlight items (proteins, phosphopeptides, and metabolites) that are significantly different, and mark out ones that are further discussed.

2. provide sources of all materials used, eg. Ti-IMAC, columns for LC-MS/MS analysis, version (date of download) of protein database used, etc.

3. the combination of 1.3 fold change and 0.3 RSD will guarantee inclusion of non-significant candidates, even worse for phosphopeptides with 1.3 fold change and 0.5 RSD. Please justify.

4. some editorial suggestions:

4a.make Venn Diagrams to scale.

4b. on page 5 line 76, I suppose you meant to say protease inhibitor cocktail?

4c. in Table 1, no scale is provided for the bars, but I prefer seeing % (in number) for easier interpretation.

6. PLOS authors have the option to publish the peer review history of their article (what does this mean?). If published, this will include your full peer review and any attached files.

Reviewer #1: No

Reviewer #2: **Yes: **Yan Wang

---

## [Author Response · Author response to Decision Letter 0]

27 Jul 2022

Dear editor,

Thank you very much for your e-mail dated 7 Jul 2022 providing with the comments of the editor and the reviewers on the manuscript entitled ‘Profiling of metabolites, proteins, and protein phosphorylation in silica-exposed BEAS-2B epithelial cells’. 

We have carefully revised our manuscript to meet the journal requirements. The raw data and associated search results were free access to iProx (Project ID: IPX0004776000, URL: https://www.iprox.cn/page/PSV023.html;?url=1658744883915Bsp2, password: dKse). Once the manuscript was accepted, the data were public. In addition, the high-resolution images were uploaded for better interpretation. 

We appreciated the comments from the reviewers, and hereby submitted our point-by-point responses.

This manuscript is original and not being considered for publication elsewhere. This study is supported by the National Natural Science Foundation of China (81900034) and Natural Science Foundation of Fujian Province (2021J01255), but the funders had no role in study design, data collection and analysis, decision to publish, or preparation of the manuscript. The authors declare no conflicts of interests.

Best Regards

Yours sincerely

Dr. Jin Chen,

The Second Affiliated Hospital of Fujian Medical University, Quanzhou, Fujian Province, China, 362000.

 

Review #1

Q1

In this manuscript, Chen et al described a multi-omics approach and results of profiling of metabolites, proteins, and protein phosphorylation in silica-exposed BEAS-2B epithelial cells, in an attempt to understand the biology of Silicosis induced by crystalline silica. Their findings showed metabolites, proteins and phosphorylation levels in many pathways were significantly altered. Many of them confirmed the findings in other similar researches.

Response: Thank you very much for your comments.

Q2

In order to deep profile the proteome and phosphopeptidome, fractionation at protein and/or peptide level is a must for mass spectrometry analysis operated in data-dependent acquisition mode. However, this study didn't use any fractionation to spread out the peptides for the mass spec to read. Therefore, the comprehensiveness of this study was largely impaired by the relatively low numbers of proteins and phosphopeptides detected.

Response: We agree with your points. Fractionation is an effective way to deep profile the protein and their phosphorylation, which could give much more information in the process with silica exposure. However, this strategy requires much more sample amount and time.

In our study, we have discovered the differential proteins and phosphorylation levels in many pathways without fractionation. Protein levels in the endoplasmic reticulum were significantly altered, and phosphorylation of MAPK signaling proteins was increased. For example, silica exposure resulted in increased phosphorylation of NF-κB, which was shown in our differential protein phosphorylation results and western blot. Furthermore, the phosphorylation of serine 727 in STAT3 was significantly increased, which is associated with regulation of transcription of inflammatory cytokines. Moreover, phosphorylation of serine 73 in JUN was enhanced, in which has been previously associated with development of silicosis (See line 311-316, page 16). That was, the performances were consistent with previous studies. Therefore, we believed our results would provide some beneficial information for further discovering the mechanism in epithelial cell induced by silica. 

Q3

Furthermore, normalization is necessary to set the base line for multiple data sets, especially for multi-omics data. This study didn’t clearly state how the data sets were normalized and cross compared. From Line 80 of Sample Prep section, it appeared that loading amounts based on protein concentrations were used to normalize for protein- and phosphopeptide- level comparison. Whereas for metabolites, internal standard (tridecanoic acid, 5 μg/mL) was added to the cells by 4:1 volume ratio (Line 96) for normalization. But knowing that cell viability varied significantly for control and treatment samples, the cell volumes and protein contents/concentration could be assumed to have greatly changed as well, therefore not a good means to normalize on.

Response: Thank you very much for your questions. Normalization was very important for data sets analysis. For protein and their phosphorylation analysis, we used the same amount proteins for quantification based on dimethyl labeling. Although cell viability significantly decreased when treated with silica (Fig. 2A), their protein level was also decreased with silica exposure. In our strategy, samples (100 µg) without and with silica treatment were labeled with light (4 μL of CH2O, 4%, v/v) and heavy (4 μL of CD2O, 4%, v/v) dimethyl isotopic reagents, respectively, and incubated with fresh catalyst 4 μL NaBH3CN (0.6 M). After incubation for 1 h at room temperature, the reaction was quenched by addition of 8 μL formic acid (5%, v/v). (See line 87-91, page 5.) In addition, we have successfully used this strategy to discover the mechanism in cancer cell migration induced by gold nanorod photothermal therapy (ACS Nano. 2018;12(9):9279-90). Moreover, the changed proteins and their phosphorylation levels were verified both in our study and previous studies, which supported our data analysis. Therefore, we used the same amount proteins in our study.

For metabolites analysis, data were normalized by internal standard and peak area (See line 138-139, page 7). Six biological replicates with 0, 24, 48 h silica exposure were used in our study, and we calculated their RSD. It was observed that the RSD of 76%, 81%, and 78% metabolites were less than 30% with 0, 24, 48 h silica incubation respectively, which indicated the repeatability and reliability of data. Therefore, we considered the method of normalization by internal standard and peak area was appropriate in our study. Besides, our results showed that phosphoenolpyruvic acid, 3-phosphoglyceric acid, 2-phosphoglyceric, and dihydroxyacetone phosphate levels increased with the increased duration of silica exposure, which indicated that silica exposure induced aerobic glycolysis in epithelial cells (See line 324-327, page 16). The performance was the same as the macrophages with silica exposure. Therefore, our metabolites data provided some useful information during the process.

To analysis the changed pathway both in metabolites and proteins, we extracted significantly differentially abundant proteins and metabolites. They were enriched in the pathway of TCA cycle, alanine, aspartate and glutamate metabolism, and pyrimidine metabolism (See line 279-281, page 14). Then we generated a figure describing the TCA cycle in detail. Increased levels of alanine, aspartic acid, N-acetylaspartic acid, ureidosuccinic acid, glutamic acid, and asparagine indicated that alanine, aspartate and glutamate metabolism were altered by silica exposure. The expression levels of the proteins PDHB, DLAT, SUCLG1, and CAD increased in response to silica exposure (See line 285-288, page 15). 

Q4

And for phosphorylation, this study only listed proteins in MAPK signaling pathway in Table 1 to show no or little protein level changes but significant changes at phosphorylation level. It would be nice to see all the phosphopeptides identified to be normalized based on the protein level changes.

Response: It is grateful of you to put forward the suggestion. We have made Table S3 in supporting information and it could provide the changes of phosphorylation levels considering their protein levels. 

(See line 231-232, page 12) We then normalized the changes of phosphorylation sites based on their protein levels to profile the alteration of phosphorylation intensity induced by silica (S3 Table).

 

Reviewer #2:

Q1

In the manuscript, the authors describe proteomics, phosphoproteomics, and metabolomics analysis of epithelial cells with and without crystalline silica exposure.

The article is well written and easy to follow. 

Response: It is very nice of you to make the comments.

Q2

Unfortunately, figures are with low resolution and I can't read details from them. Raw data need to be provided for evaluation too, which is missing. With this I can't make a responsible review of the presented manuscript.

Response: We apologize for the low-resolution images and the missing of raw data. Image was replaced by high-resolution ones. The raw data and associated search results were free access to iProx. (URL: https://www.iprox.cn/page/PSV023.html;?url=1658744883915Bsp2, password: dKse).. Once the manuscript was accepted, the data were public.

(See line 135-137, page 7) The raw data and associated search results were free access to iProx (Project ID: IPX0004776000).

Q3

Here are some suggestions from what I can read:

1. in supplemental table, highlight items (proteins, phosphopeptides, and metabolites) that are significantly different, and mark out ones that are further discussed.

2. provide sources of all materials used, eg. Ti-IMAC, columns for LC-MS/MS analysis, version (date of download) of protein database used, etc.

3. the combination of 1.3 fold change and 0.3 RSD will guarantee inclusion of non-significant candidates, even worse for phosphopeptides with 1.3 fold change and 0.5 RSD. Please justify.

Response: We really appreciate the reviewer’s suggestion.

1. We made some changes in our supplemental tables. A column was added in tables to label the items with significant change, and those which were further discussed were also labeled. 

2. The sources of materials were refreshed. 

(See line 93-94, page 5) Phosphopeptides were enriched using Ti-IMAC (J&K Scientific, China) as we described previously.

(See line 117-119, page 6-7) Peptides and phosphopeptides were loaded onto a C18 trap column and separated using an 18 cm× 75 μm i.d. C18 (1.9 μm, 120 Å, purchased from Dr. Maisch, Germany) capillary column at a flow rate of 300 nL/min.

(See line 125-128, page 7) MS data were processed using MaxQuant (Version 1.5.8.3) using Andromeda as a search engine against the Uniprot human protein database (Download from uniport with Proteome ID UP000005640) with precursor mass tolerance of 4.5 ppm and fragment mass deviation of 0.5 Da.

3. Since phosphorylation is a dynamic process, we increased the RSD of three biological replicates to 0.5 for discovering significant candidates. In our study, phosphoproteome analysis showed that the MAPK signaling pathway was significantly affected by silica treatment (See line 233-234, page 12). It showed that silica exposure resulted in increased phosphorylation of NF-κB, which was indicative of activation and subsequent inflammation. Furthermore, the phosphorylation of serine 727 in STAT3 was significantly increased, which is associated with regulation of transcription of inflammatory cytokines. Moreover, phosphorylation of serine 73 in JUN was enhanced, in which has been previously associated with development of silicosis (See line 311-316, page 16). Therefore, our data process could provide beneficial information.

Q4

4. some editorial suggestions:

4a. make Venn Diagrams to scale.

4b. on page 5 line 76, I suppose you meant to say protease inhibitor cocktail?

4c. in Table 1, no scale is provided for the bars, but I prefer seeing % (in number) for easier interpretation.

Response: Thank you very much for your editorial suggestion.

4a. Venn Diagrams were refreshed for better understanding (See Figure 3A and 4A).

4b. The words were corrected for avoiding misunderstanding.

(See line 79-81, page 5) The cells were lysed in a buffer containing 50 mM HEPES (pH 7.4), 4% SDS, protease inhibitor cocktail (Sigma, U.S.A.) and phosphatase inhibitors (Beyotime, China).

4c. The number (%) was added in Table 1 for easier interpretation (See Table 1).

---

## [Decision Letter · Decision Letter 1]

2 Aug 2022

Profiling of metabolites, proteins, and protein phosphorylation in silica-exposed BEAS-2B epithelial cells

PONE-D-22-15393R1

Dear Dr. Chen,

We’re pleased to inform you that your manuscript has been judged scientifically suitable for publication and will be formally accepted for publication once it meets all outstanding technical requirements.

Kind regards,

Aleksandra Nita-Lazar, Ph. D.

Academic Editor

PLOS ONE

Additional Editor Comments (optional):

Reviewers' comments:

Reviewer's Responses to Questions

**Comments to the Author**

1. If the authors have adequately addressed your comments raised in a previous round of review and you feel that this manuscript is now acceptable for publication, you may indicate that here to bypass the “Comments to the Author” section, enter your conflict of interest statement in the “Confidential to Editor” section, and submit your "Accept" recommendation.

Reviewer #1: All comments have been addressed

2. Is the manuscript technically sound, and do the data support the conclusions?

Reviewer #1: Partly

3. Has the statistical analysis been performed appropriately and rigorously? 

Reviewer #1: Yes

4. Have the authors made all data underlying the findings in their manuscript fully available?

Reviewer #1: Yes

5. Is the manuscript presented in an intelligible fashion and written in standard English?

Reviewer #1: Yes

6. Review Comments to the Author

Reviewer #1: (No Response)

7. PLOS authors have the option to publish the peer review history of their article (what does this mean?). If published, this will include your full peer review and any attached files.

Reviewer #1: No

---

## [Editor Report · Acceptance letter]

5 Sep 2022

PONE-D-22-15393R1 

Profiling of metabolites, proteins, and protein phosphorylation in silica-exposed BEAS-2B epithelial cells 

Dear Dr. Chen:

I'm pleased to inform you that your manuscript has been deemed suitable for publication in PLOS ONE. Congratulations! Your manuscript is now with our production department. 

Kind regards, 

on behalf of

Dr. Aleksandra Nita-Lazar 

Academic Editor

PLOS ONE